# Post-Translational Modifications Evoked by Reactive Carbonyl Species in Ultraviolet-A-Exposed Skin: Implication in Fibroblast Senescence and Skin Photoaging

**DOI:** 10.3390/antiox11112281

**Published:** 2022-11-18

**Authors:** Anne Negre-Salvayre, Robert Salvayre

**Affiliations:** Faculty of Medicine, Department of Biochemistry, INSERM U1297 and University of Toulouse, 31432 Toulouse, France

**Keywords:** skin photoaging, reactive carbonyl species, carbonyl stress, fibroblast senescence, actinic elastosis

## Abstract

Photoaging is an accelerated form of aging resulting from skin exposure to ultraviolet (UV) radiation. UV-A radiation deeply penetrates the dermis and triggers the generation of reactive oxygen species (ROS) which promotes damage to DNA, lipids and proteins. Lipid peroxidation results from the oxidative attack of polyunsaturated fatty acids which generate a huge amount of lipid peroxidation products, among them reactive carbonyl species (RCS) such as α, β-unsaturated hydroxyalkenals (e.g., 4-hydroxynonenal), acrolein or malondialdehyde. These highly reactive agents form adducts on free NH2 groups and thiol residues on amino acids in proteins and can also modify DNA and phospholipids. The accumulation of RCS-adducts leads to carbonyl stress characterized by progressive cellular and tissular dysfunction, inflammation and toxicity. RCS-adducts are formed in the dermis of skin exposed to UV-A radiation. Several RCS targets have been identified in the dermis, such as collagen and elastin in the extracellular matrix, whose modification could contribute to actinic elastosis lesions. RCS-adducts may play a role in fibroblast senescence via the modification of histones, and the sirtuin SIRT1, leading to an accumulation of acetylated proteins. The cytoskeleton protein vimentin is modified by RCS, which could impair fibroblast motility. A better identification of protein modification and carbonyl stress in the dermis may help to develop new treatment approaches for preventing photoaging.

## 1. Introduction

Aging is a multifactorial process characterized by a sequential and progressive decline of physiological and biological functions, a decreased ability to adapt to metabolic stresses and an increased risk of developing diseases. This is dependent on intrinsic and extrinsic processes as well as genetic and environmental factors [1,2]. Photoaging is an accelerated form of chronological skin aging caused by repeated exposure to solar or artificial ultraviolet (UV) radiation [3,4]. Photoaging is characterized by an alteration of skin barrier properties, formation of deep wrinkles, loss of skin tone, and formation of spider veins and pigmented spots [3,5,6]. Connective tissue in the dermis represents a main target of UV radiation, with the formation of lesions such as actinic elastosis, a hallmark of photoaging [7,8]. Actinic elastosis results from the accumulation of elastotic material composed of abnormal disorganized elastin and fibrillin with an important loss of fibrillar collagen and proteoglycans [4,9,10,11].

A main feature of both skin photoaging and chronological aging is the senescence of dermis fibroblasts [12,13]. These cells are the primary source of extracellular matrix components that maintain the structural and mechanical properties of the skin [13]. Dermis fibroblasts continuously undergo photoaging damage which progressively alters their structure and function by promoting telomere shortening, gene expression changes, cell cycle arrest, cytoskeleton alterations and regeneration impairment. All these events combine to produce a senescence-associated secretory phenotype (SASP) characteristic of senescent cells [12,13].

The generation of reactive oxygen species (ROS) by UV radiation is a key-feature of the photoaging process [4,5,14]. UV radiation produces ROS and progressively alters the cellular defense mechanisms (melanin, degradation and repair processes, antioxidant systems) [15,16], leading to oxidative stress and subsequent skin damage [17,18,19]. Carbonyl stress is a consequence of oxidative stress and is characterized by the formation of reactive carbonyl groups leading to the formation of advanced glycation end products (AGEs) and advanced lipid oxidation end products (ALEs), a hallmark of both chronological aging and photoaging [20,21,22,23]. Reactive carbonyl species (RCS) issued from the oxidation of polyunsaturated fatty acids (PUFAs) are strong alkylating agents which form adducts on free amino (NH2) groups and thiol (SH) residues on proteins and other macromolecules, causing conformational changes and progressive cell dysfunction [24,25,26,27]. RCS-adducts are abundantly present in pathologies associated with oxidative stress such as atherosclerosis and cardiovascular diseases, inflammatory and neurodegenerative diseases, and cancers [24,25]. RCS-adducts are detected in the epidermis and dermis of skin exposed to UV radiation. This review is focused on the post-translational modifications exerted by the most-studied RCS (4-hydroxynonenal or HNE, malondialdehyde or MDA, acrolein) in the dermis of skin exposed to UV-A and their possible implications in the photoaging process.

## 2. UV-Induced ROS Production and Photoaging

The term “photoaging” specifically refers to changes and modifications of the skin resulting from long-term repeated exposure to UV radiation from solar or artificial UV sources [5,28,29], and mostly visible on sun-exposed areas, i.e., face, neck, upper chest, hands, forearms or lips [29,30,31]. 

Solar UV-A radiation is responsible for more than 95% of skin-related effects, which represents the main source of ROS production in the dermis [17,32,33]. These ROS result from interactions between UV photons and endogenous photosensitizers (porphyrins, riboflavin, quinines…), that become excited and react with O_2_ to produce singlet oxygen, OH°, O_2_^•−^, or H_2_O_2_ [34]. ROS stimulate various signaling responses and redox-sensitive transcription factors in fibroblasts, triggering inflammation, proliferation, senescence or apoptosis, expression and secretion of metalloproteases, collagenases and gelatinase, implicated in ECM degradation, collagen loss and elastin damage. ROS indirectly provoke DNA oxidation and mutations evidenced by the formation of 8 oxo-deoxyguanosine (8-oxo-dG) in guanine bases, and are involved in skin cancer development [34,35]. The direct implications of ROS in photoaging have been thoroughly investigated (for review, see [31,36,37,38]). Lipid oxidation products generated from PUFA oxidation by UV radiation largely contribute to the photoaging process, particularly aldehydic RCS (HNE, MDA, acrolein) [22,24,25,39,40,41,42,43].

## 3. Mechanisms Leading to RCS Generation and Adduct Formation

ROS and oxidative stress promote the oxidation of PUFAs, generating a huge amount of highly reactive lipid oxidation products, mainly lipid hydroperoxides (LOOH) and numerous aldehydes (RCS), including MDA, acrolein and HNE [22,23,39,44,45]. These agents rapidly react with free NH2 groups and SH residues on proteins forming Schiff bases and stable Michael adducts [45,46,47]. The main amino-acid targets are lysine (Lys), the imidazole group on histidine (His), the guanidine group on arginine (Arg), or the thiol group of cysteine (Cys) [45,46,47]. The formation and nature of RCS-adducts have been extensively reviewed by Domingues et al. [46]. RCS-adduct formation increases with aging and in pathologies associated with oxidative stress and lipid peroxidation [47]. 

HNE is an α, β-unsaturated hydroxy-alkenal formed during the oxidation of n-6 PUFAs (linoleic and arachidonic acids) [22]. Other α, β-unsaturated aldehydes are also produced from the oxidation of n-3 PUFAs (docosahexaenoic acid, eicosapentaenoic acid, linolenic acid), such as 4-hydroxyhexenal (4-HHE) and 4-oxo-trans-2-nonenal (ONE) [48]. HNE reacts with Cys, His and Lys residues on proteins to form stable Michael adducts with a hemiacetal structure [26]. HNE may modify DNA on deoxyguanosine [49], and forms adducts on phosphatidylethanolamine which could alter the properties of cellular membranes [50,51]. HNE-adducts are easily formed on signal transduction proteins, such as growth factor receptors, kinases, phosphatases, lipoproteins, transcription factors, and mitochondrial components, leading to cellular dysfunction and toxicity [47,48,52]. The formation of HNE-adducts increases with aging and in age-related pathologies together with increased oxidative stress and high-lipid-peroxidation levels [48]. 

MDA is abundantly produced via the oxidation of n-3 and n-6 PUFAs [22]. It is considered a reliable marker of lipid peroxidation in tissues and circulating fluids [53]. MDA reacts with Lys residues on proteins to form Schiff bases, particularly on lipoproteins, with strong implication in their metabolic deviation towards macrophages and the formation of foam cells [22,47,48]. MDA adducts accumulate in lipofuscin, a fluorescent pigment containing oxidized proteins, metals and sugars. The presence of lipofuscin is a hallmark of tissular aging [54]. Acrolein (CH2 = CH–CHO) is the shortest unsaturated lipid-peroxidation-derived aldehyde. Acrolein reacts with Cys, His and Lys nucleophile residues [45] and is involved in the formation of protein cross-links [55]. 

RCS such as α-oxoaldehydes (methylglyoxal and glyoxal) are basically generated from the Maillard reaction during diabetes, via the condensation of sugars with proteins. These agents are also generated from lipid peroxidation and react with Lys and Arg residues to form AGEs [56,57]. AGEs contribute to the pathophysiology of duabetes and accumulate during physiological aging, particularly in the skin [20]. 

The possible role of RCS and carbonyl stress in photoaging is summarized in Figure 1.

## 4. Post-Translational Modifications of Dermis Components by RCS

As recently reviewed by Bernerd et al. [37] and Papaccio et al. [58], chronic UV exposure promotes protein oxidation and carbonylation in the epidermis and dermis, and the accumulation of oxidatively modified and damaged proteins. RCS-adducts are detected in human skin biopsies, in keratinocytes, fibroblasts [59,60] and ECM [61]. Different MDA-derived epitopes can be detected in cutaneous superficial spreading melanoma and squamous cell carcinoma [59]. These MDA-epitopes are present in the age-related pigment lipofuscin and can behave as photosensitizers of UV-A-induced oxidative stress [62]. *In vitro*, RCS (HNE, acrolein) stimulate inflammatory signaling pathways in cultured skin fibroblasts leading to their premature senescence [63,64,65,66,67]. So far, a few proteins directly or indirectly targeted by RCS have been identified with a possible role in skin aging. Le Boulch et al. and Baraibar et al. [65,68] identified the “oxi-proteome” (carbonylated proteins) of fibroblasts undergoing oxidative-stress-induced premature senescence. The most important modifications were observed on proteins of the cytoskeleton and energy metabolism, including vimentin, glucose-6-phosphate dehydrogenase and the heat shock 71 kDa protein. Likewise, the proteomic analysis of UV-A-exposed murine skin showed that UV-A treatment altered protein systems on calcium signaling, mitochondrial function or sirtuin expression [69]. In this study, the protective effect of carnosine, an efficient carbonyl scavenger capable of neutralizing HNE-adducts [70], suggested that UV-induced proteome alterations may result from RCS [69]. In the dermis of UV-exposed skin, RCS-adducts are detected on ECM and in fibroblasts. 

### 4.1. Post-Translational Modifications Elicited by RCS in ECM 

#### 4.1.1. Collagen

ECM components (collagen, elastin, glycosaminoglycans), are primary targets of oxidative and carbonyl attack during photoaging [58]. Zucchi et al. recently reported that carbonylated proteins are formed on the connective tissue in the dermis from sun-exposed human skin, specifically MDA-adduct deposits, which may accumulate on collagen [44]. The formation of MDA-adducts on collagen could be observed in vitro on acellular dermis models in the presence of MDA [44], in agreement with previously reported data showing that MDA may modify collagen [71]. Interestingly, the formation and accumulation of MDA-adducts on collagen was associated with a yellowing of the acellular dermis, suggesting a role for protein carbonylation in the yellowish change of the dermis occurring during skin photoaging [44] and in actinic elastosis development [41]. The mechanism by which these MDA-adducts contribute to the dermis yellowing still remains unknown [44]. 

#### 4.1.2. Elastin

Actinic elastosis results from the accumulation of elastotic material, i.e., abnormal disorganized elastin and fibrillin, a microfibrillar component of elastic fibers, with an important loss of fibrillar collagens and proteoglycans [4,9,10,11]. Histological analysis of actinic elastotic lesions using haematoxylin/eosin stain, revealed the presence of basophilic “spaghetti-like” curled fibers, a hallmark of elastotic material. The characterization of constituents in immunofluorescence studies indicates that elastin abundantly accumulates in the elastotic material, together with fibrillin and fibronectin, and to a lesser extent, type I and type VI collagen, glycosaminoglycans and lysozyme [10,72]. 

The origin of elastotic material in the skin is not clear. A neosynthesis of elastin has been suggested [10]. Indeed, there is an increased expression of elastin and fibrillin in sun-exposed skin [73], while the exposure of dermal fibroblasts to UV-B radiation [73] or ROS generated by the xanthine/xanthine oxidase system [74], triggers an increase in tropoelastin mRNA. 

RCS (HNE, acrolein) inhibit the elastogenic activity of TGFβ1 in vitro in dermal fibroblasts, via a post-translational modification of the EGF receptor [64]. This modification activates an EGFR-signaling which antagonizes the stabilization of tropoelastin mRNA evoked by TGFβ1 [75]. The accumulation of elastin in elastotic material could result from a defective degradation by neutrophil elastase. Various mechanisms could be involved, such as the expression of elafin, an elastase inhibitor [76], or a post-translational modification of elastin impairing its degradability by elastase. Schalkwijk et al. reported a modification of elastin via a transglutaminase-mediated cross-linking mechanism between elastin and other proteins [77]. Another proposed mechanism is the accumulation of AGEs on elastin, particularly Nε-(carboxymethyl) lysines which inhibit elastase activity [78]. 

HNE- and acrolein-adducts accumulate on elastin in actinic elastosis lesions [61] and could contribute to the accumulation of elastotic material, as observed in hairless mice exposed to UV-A [42]. It can be hypothesized that HNE- and acrolein-adducts impair the degradation of elastin by elastase [42]. It is worth noting that elastin is not modified by HNE or acrolein in arteries during the process of intrinsic aging, suggesting that the presence of RCS-adducts on elastin is specific to photoaging [79]. The mechanism by which RCS modify elastin is not known. Using transmission electronic microscopy, Dhital et al. reported that UV-A exposure elicits fragmentation of elastic fibers and a decrease in desmosine content [80], possibly associated with desmosine photolysis and a release of free lysine [81]. One hypothesis is that elastin fragmentation elicited by UV-A may expose the free amino group of lysine allowing the formation of RCS-adducts and elastin modification.

### 4.2. Post-Translational Modifications Evoked by RCS in Dermal Fibroblasts. Implications in Fibroblast Senescence

#### 4.2.1. Fibroblast Senescence in Skin Photoaging

Dermal fibroblasts produce ECM components (collagen, fibronectin, glycosaminoglycans…), that help to maintain skin elasticity and moisture [82]. During long-term or repeated sun exposure, dermal fibroblasts are continuously subjected to UV-induced ROS which progressively promote phenotypic changes, DNA damage, chromosome instability, telomere shortening, and cell cycle arrest via an increased expression of the cell cycle kinase inhibitors p16 or p21, with all these events leading to cell dysfunction and skin aging [12,13,83,84,85]. Upon UV exposure, fibroblasts exhibit morphological changes, and become larger and flattened with altered gene expression [86]. These cells progressively acquire a senescence-associated secretory phenotype (SASP), characterized by decreased cell proliferation, an increased secretion of MMPs and the degradation of ECM. Several senescence biomarkers could be observed including increased activity of the senescence-associated β-galactosidase (SA-βGal) [87], or the accumulation of γH2AX foci in their nuclei, indicative of DNA strand break formation [88]. Alterations of the cytoskeleton are observed, particularly on vimentin, a cytoskeletal protein possibly modified by glycoxidation and by RCS [67,89,90]. The SASP phenotype is associated with mitochondrial dysfunction and an increased production of mitochondrial ROS, as supported by experiments showing that mice conditionally deficient for mitochondrial superoxide dismutase (SOD2) in fibroblasts, undergo accelerated senescence [91]. Mitochondrial dysfunction promotes the activation of redox-sensitive transcription factors NF-κB and AP-1, and a chronic inflammatory environment in senescent fibroblasts [92,93]. The SASP phenotype could be associated with decreased activity of the proteasome system and the accumulation of oxidized and modified (carbonylated) proteins [94,95]. Note that the premature senescence of human fibroblasts and keratinocytes exposed to UV-B is not associated with proteasome inhibition [96,97].

#### 4.2.2. DNA Alterations by Aldehydes in Senescent Fibroblasts

The accumulation of RCS (MDA, HNE, ONE, acrolein) in fibroblasts exert anti-proliferative, genotoxic and cytotoxic effects possibly associated with the presence of adducts on DNA (modification of deoxyguanosine) or the formation of DNA crosslinks with proteins [27,66,98]. HNE promotes the modification of histones (particularly histone H2A) [99], which affects their conformation and their ability to bind DNA. Moreover, HNE could modify and inactivate histone deacetylases HDAC, leading to changes in histone acetylation patterns and transcription of HDAC-repressed genes [100]. HNE and RCS directly elicit DNA damage evidenced by the accumulation of γH2AX foci in skin fibroblasts exposed to UV-A radiation or incubated with HNE [67]. Similarly, an increased content of γH2AX foci is observed in the skin of UV-A-irradiated hairless mice [67]. Interestingly, a daily topical application of the carbonyl (HNE) scavenger carnosine on UV-A-irradiated animals decreased the γH2AX foci content together with a decreased expression of HNE adducts, indicating that neutralization of HNE may protect against DNA damage evoked by UV-radiation in the skin [42,54].

#### 4.2.3. SIRT1 Modifications in Senescent Fibroblasts 

Oxidative stress and RCS exert a dual effect on sirtuin activity (silent information regulator proteins), a family of nicotinamide adenine dinucleotide (NAD)-dependent histone deacetylases, which catalyze the deacetylation of histones and non-histone proteins [101]. Among sirtuins, SIRT1 has been widely investigated and regulates several biological pathways including energetic state and genome integrity, DNA repair, cellular senescence and lifespan extension. The mechanisms of SIRT1 regulation involve the deacetylation of several systems such as p53, forkhead-box transcription factor (FoxO3α), NF-κB and histones [102,103,104]. SIRT1 overexpression in skin fibroblasts could protect against UV-B-induced damage by deacetylating FoXO3α and by increasing cellular resistance to oxidative stress [105]. SIRT1 could reverse p53 acetylation and cell cycle arrest evoked by UV-B [105]. In contrast, under high oxidative stress and chronic inflammatory conditions, there is a decrease in SIRT1 expression possibly associated with its post-translational modification by HNE or acrolein. Highly modified SIRT1 is inactivated and degraded by proteasome machinery, leading to the accumulation of acetylated substrates which aggravates the inflammatory and prosenescent cellular responses [106]. SIRT1 activity is altered in skin fibroblasts exposed either to UV-A radiation or to HNE in correlation with the installation of the SASP phenotype [67]. The role of HNE in SIRT1 dysfunction was supported by the proteomic analysis of UV-A-exposed murine skin, which showed alterations of sirtuin expression that were restored by pretreatment with the HNE-scavenger carnosine [69]. 

#### 4.2.4. Vimentin Modification in Senescent Fibroblasts

UV-A-exposed fibroblasts exhibit morphological changes and cytoskeleton alterations such as inhibition of actin filament polymerization, reduction in actin filament number and upregulation of the capping protein muscle Z-line α1 (CAPZA1), an actin filament polymerization inhibitor which promotes a reduction in collagen synthesis [107]. 

An alteration of the cytoskeletal structure is observed with HNE which could interact with microtubules and microfilaments, causing depolymerization of the microtubular structures, dissolution of stress-fibers and an impairment in cell motility, possibly via interactions with vimentin [108]. Indeed, microtubule components, particularly the intermediate filament vimentin, control cell motility, migration and wound-healing in fibroblasts by regulating actomyosin contraction forces and interactions with ECM [109,110]. 

As reported by Monico et al. [111], vimentin is a target of oxidative stress and RCS (HNE), via the modification of the cysteine residue (Cys328), resulting in the disruption of the intermediate filament network and the generation of intracellular aggresomes [112]. Vimentin modification alters the motility of fibroblasts by triggering a loss in contractile capacity as observed in senescent cells [113]. The presence of HNE-modified vimentin has been detected in the skin of UV-A-irradiated hairless mice, and in UV-A or HNE-exposed dermal fibroblasts (Figure 2), together with the development of the SASP phenotype evoked by these treatments [67]. HNE-modified vimentin could be detected at the cell surface in fibroblasts, in agreement with Frescas et al. [90], who suggested that vimentin modification by RCS (at cell surface), constitutes an “eat me” signal allowing the phagocytosis of senescent cells. This mechanism is progressively impaired in aging cells [90]. Note that vimentin is also a target of glycation and AGEs, which promote its accumulation in aggresomes [89]. 

## 5. Prevention of Carbonyl Stress to Limit Skin Photoaging

Several endogenous biological systems protect the skin from the deleterious effect of UV light and subsequent oxidative and carbonyl stresses. The first mechanism of defense is constituted by melanin, mainly eumelanin in the skin, a natural pigment of skin color resulting from tyrosine oxidation and polymerization in melanocytes. Eumelanin may absorb light and prevents UV-A and UV-B penetration, thereby protecting cells from UV-radiation damage [1,2,3,4,5,6]. 

The second line of defense is constituted by antioxidant compounds and systems that protect the skin against UV-induced oxidative and carbonyl stresses. Classical antioxidant compounds, either endogenous (glutathione) or exogenous agents such as vitamins E and C, carotenoids (e.g., carotenes, lycopene, alloxanthin, lutein, zeaxanthin) act by scavenging ROS. Topical application of antioxidants (vitamin C, vitamin E, beta-carotenes, polyphenols…) protects the skin against the effects of UV-radiation, depending on their capacity to penetrate the skin barrier. These agents could be incorporated to broad-spectrum sunscreens whose topical application is necessary to absorb UV-A and -B and assure skin protection [15]. 

Various endogenous antioxidant enzymatic systems are present in the skin. Antioxidant enzymes, such as superoxide dismutases, catalase, glutathione peroxidase, act by metabolizing ROS, whereas another set of redox enzymes, such as glutathione reductase, thioredoxin, thioredoxin reductase, maintain the reductive potential of the cytoplasm [16,114]. In addition, redox-sensitive transcription factors, such as the nuclear factor erythroid 2 related factor 2 (Nrf2), enhance cellular antioxidant defenses in response to stress evoked by UV-A radiation in skin fibroblasts [115]. 

Antioxidants are unable to inhibit the formation of RCS once these agents are formed. Several endogenous systems may neutralize or metabolize RCS and prevent carbonyl stress. Glutathione (GSH) is a potent biological carbonyl scavenger, particularly active in HNE via the formation of HNE-GSH adducts, which neutralize HNE bioreactivity and inhibit the formation of HNE-protein adducts [116]. The reaction is catalyzed by glutathione-S-transferase (GST), particularly GST A4-4 which is insensitive to HNE-adduction in contrast to GST A1-1 and is catalytically more active than the other GSTs [117]. Other systems involved in HNE neutralization are the alcohol dehydrogenases (ADH) and aldehyde dehydrogenases (ALDH), which catalyze HNE metabolism into 4-hydroxynonenoic acid (HNA) and alcohol 1,4-dihydroxynonene (DHN), respectively [118]. 

A few agents with carbonyl-scavenging properties could neutralize carbonyl stress in skin cells, particularly N-acetyl cysteine and carnosine. N-acetyl cysteine and may increase the GSH content [119]. When added to skin fibroblasts, N-acetyl cysteine enhances the expression of ECM genes [120]. *In vivo*, topical pretreatment with N-acetyl cysteine protects against skin-aging evoked by high or low UV-A radiation in hairless mice [121]. Likewise, the cysteine derivative 3,3-dimethyl-D-cysteine and D-penicillamine are two potent carbonyl scavenger agents which protect fibroblasts against glyoxal or methylglyoxal toxicity [122]. 

L-carnosine (β-alanyl-L-histidine) exhibits both antioxidant and carbonyl scavenger properties preventing the formation of AGEs and RCS-adducts on proteins. L-carnosine is an anti-inflammatory, metal-chelating and antioxidant molecule [70], with potential protective effects against cardiovascular [123], metabolic [124] and neurodegenerative diseases [125], cancer [126], and aging [127]. L-carnosine exerts several antiaging properties in skin fibroblasts by inhibiting the telomere-shortening process [128] and by improving their replicative lifespan [129]. Recent studies reported that L-carnosine could prevent or delay fibroblast senescence evoked by UV-A radiation or by HNE [67]. More precisely, L-carnosine prevented the modification of SIRT1 and vimentin by HNE and the accumulation of acetylated and ubiquitinated proteins which characterize the fibroblast-aging process. *In vivo*, the topical application of L-carnosine on the skin of UV-A-treated hairless mice, restored a normal proteomic profile [69]. L-carnosine reduced fibroblast senescence markers and actinic elastosis lesions, in correlation with a decreased accumulation of HNE-adducts on dermal proteins including elastin and vimentin [42,67]. New histidine-containing dipeptides would be designed or developed from the parent L-carnosine structure, to ameliorate the carbonyl scavenger activity, with better efficacy than carnosine (better stability, higher quenching activity). Among these agents, Cyclo (His-Pro) (CHP) is a dietary-cyclized His-Pro dipeptide capable of forming two symmetric dipeptides (His-Pro, Pro-His) in an acidic environment. CHP is more stable than L-carnosine, with greater anti-inflammatory, antioxidant and carbonyl-scavenging activity [130]. Whether and how CHP could prevent the formation of RCS-adducts in the skin remains to be investigated. 

## 6. Conclusions and Perspectives

Lipid peroxidation and carbonyl stress are involved in the process of skin photoaging resulting from skin exposure to UV-radiation. Aldehydic RCS (HNE, acrolein, MDA…) are formed during the peroxidation of PUFAs and contribute to dermal photoaging by forming adducts on several protein targets in ECM (collagen, elastin) and fibroblasts, in turn promoting their senescence (Figure 3). 

RCS may directly stimulate senescent signaling pathways in fibroblasts including telomere shortening and the modification of SIRT1 and vimentin. Based on proteomic studies of dermal fibroblasts exposed to UV-radiation, many other systems would be altered by RCS with consequences in skin photoaging [65,68]. For instance, mitochondria components in skin cells could be altered by aldehydes issued from cardiolipin oxidation, as reported in cancer [131], with possible consequences for bioenergetics, mitophagy and ROS-production in the dermis.

It would be interesting to investigate whether carbonyl scavengers either dietary or incorporated into sunscreen preparations for topical application may minimize or delay the development of skin photoaging, as reported with antioxidants [15]. In this context, topical application of N acetyl cysteine and carnosine efficiently protected the skin of hairless mice against the development of deep wrinkles, actinic elastosis lesions and fibroblast senescence evoked by UV-A radiation, via (in part) an inhibition of RCS-adduct accumulation [42,67,121]. The design and development of new molecules with greater carbonyl-scavenger properties associated with better stability, tolerance and resistance to degradation, may open new perspectives for skin protection against photoaging and further applications.

## Figures and Tables

**Figure 1 antioxidants-11-02281-f001:**
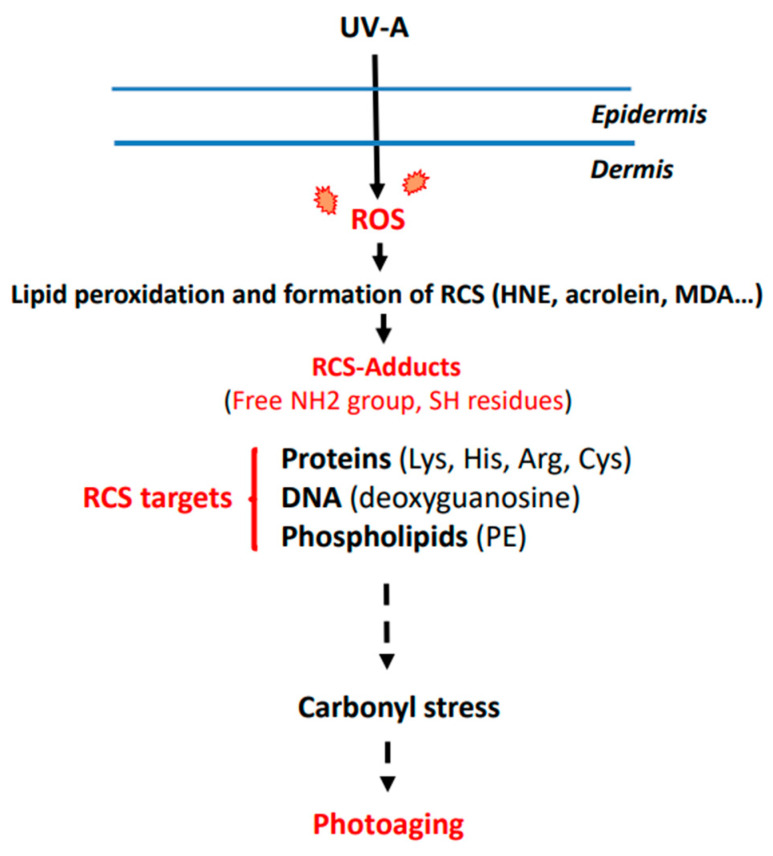
Hypothetical mechanism of carbonyl stress in skin exposed to UV-A radiation. Lys, lysine; His, histidine; Arg, arginine; Cys, cysteine; PE, phosphatidylethanolamine.

**Figure 2 antioxidants-11-02281-f002:**
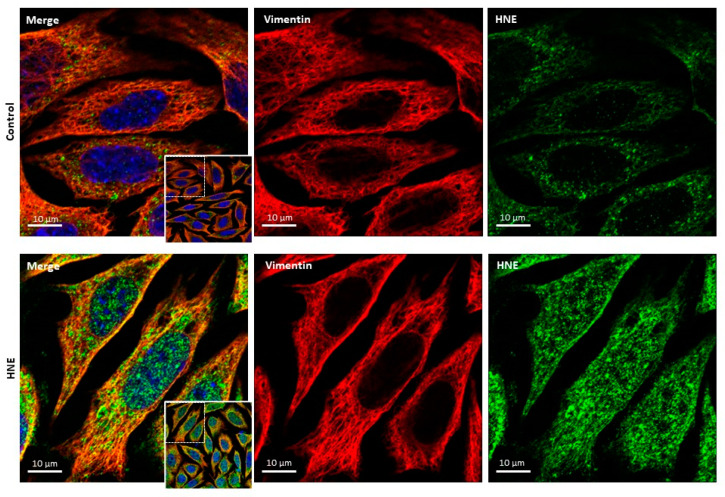
Formation of HNE-adducts on vimentin in murine skin fibroblasts. Cultured murine fibroblasts were exposed to HNE (30 µM, 4 h in HBSS medium). At the end, cells were fixed in paraformaldehyde (4% in phosphate buffered saline, PBS, for 10 min), and then used for confocal imaging studies, in the previously reported conditions [67]. After blocking with PBS containing 5% bovine serum albumin for 45 min, cells were incubated with the primary anti-HNE-Michael adduct antibody (#MA5-27570, Invitrogen), or the anti-vimentin antibody (#92547, Abcam), followed by Alexa Fluor-488 or Alexa Fluor-546 conjugated antibodies. Representative confocal imaging pictures show the presence of HNE-adducts (green, **right** panel) on vimentin (red, **middle** panel), and the merge (yellow, **left** panel), in fibroblasts incubated with HNE (lower pictures) vs. control without HNE (upper pictures). Scale bar = 5 µm. Original 63× confocal images are shown inset.

**Figure 3 antioxidants-11-02281-f003:**
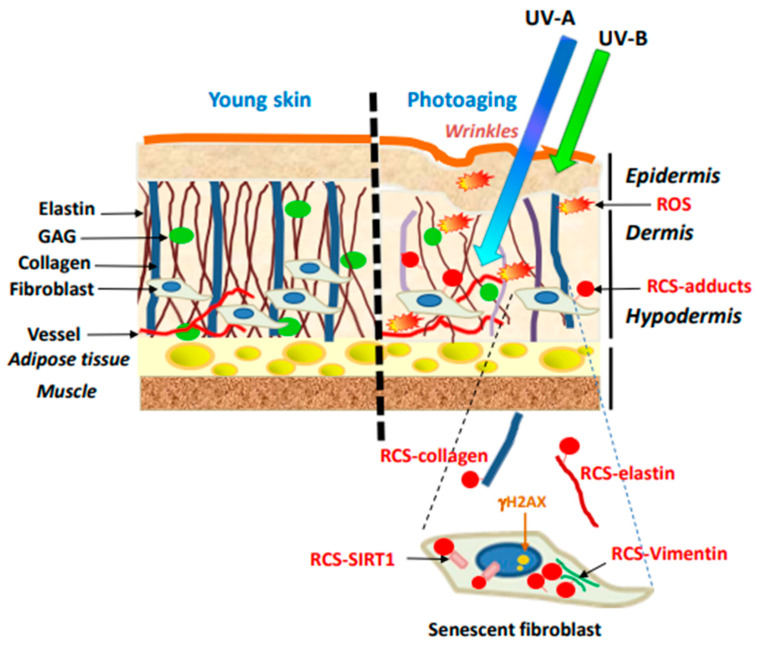
Schematic representation of RCS effects in skin photoaging. (**Left** panel), young skin; (**Right** panel), photoaging. GAG, glycosaminoglycans; ROS, reactive oxygen species; RCS, reactive carbonyl species.

## Data Availability

Not applicable.

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
