# Peer review of "Post-Translational Modifications Evoked by Reactive Carbonyl Species in Ultraviolet-A-Exposed Skin: Implication in Fibroblast Senescence and Skin Photoaging"

_antioxidants, 2022, doi:10.3390/antiox11112281_

Round 1
Reviewer 1 Report
This study focused on the photoaging activity caused by Ultra-Violet-A radiation (UVA). UVA-exposed hairless mice exhibited reactive carbonyl species (RCS) -modified proteins including collagen and elastin in the extracellular matrix and accelerated fibroblast senescence. RCS modified DNA and histones, the sirtuin (SIRT1) and the cytoskeleton protein vimentin stimulating fibroblast senescence. Some targets of RCS in the dermis and their contribution to dermal photoaging was discussed in this manuscript.
Comments
1. Most of the information present in ”UV-induced ROS production and photoaging changes” section was well-known. For example, the wavelength of UV-A, -B and -C and UV caused DNA damage. Some new information must present in this section.
2. The mechanisms and regulation of RCS-induced protein expression and signal transduction causing photoaging have to describe clearly in the manuscript.
3. Figures present the signal regulation of RCS-induced photoaging may help the readers to understanding the mechanisms involved in this issue.
4. The authors stated that “We discuss the efficacy of agents with carbonyl scavenger activity, able to neutralize or inhibit the biological reactivity of RCS and their potential interest to prevent skin photoaging” in Abstract section. However, the efficacy of agents with carbonyl scavenger activity did not present in the manuscript clearly.
5. The abbreviation must define at the first time present in the manuscript.
6. Some typing errors were in the text. The manuscript must recheck carefully.
Reviewer 2 Report
Ms-antioxidants-2005976 can be accepted after correcting some typos present in the text
Author Response
We thank the reviewer for his/her encouraging comments
The manuscript has been carefully proofread and most corrections have been tentatively taken into account.

Reviewer 3 Report
Article titled 'Post-translational modifications evoked by reactive carbonyl species in Ultra-Violet-A-exposed skins: Implication in fibro blast senescence and skin photoaging' could be an interesting contribution to the journal.
However, a few aspects need to be addressed beforehand in the benefit of the reader:
It is a bit unclear what this review brings as elements of novelty, as there are already lots of reviews on this particular topic in the medical literature in this field. Moreoover, the review is a bit to concise and could benefit from being more elaborate.
Grammar and punctuation should be verified throughout the entire article.
The last sentence in "We discuss the efficacy of agents with carbonyl scavenger activity, able to neutralize or inhibit the biological reactivity of RCS and their potential interest to prevent skin photoaging." could be rephrased.
Round 2
Reviewer 1 Report
Authors have response their manuscript according to the comments. This manuscript is acceptable for publishing.
Author Response
Thank you.
Reviewer 3 Report
Manuscript has been improved and could be considered for publishing if the Editor accepts.
Author Response
Thank you.